# Spontaneous Non-Aneurysmal Convexity Subarachnoid Hemorrhage: A Scoping Review of Different Etiologies beyond Cerebral Amyloid Angiopathy

**DOI:** 10.3390/jcm13154382

**Published:** 2024-07-26

**Authors:** Marialuisa Zedde, Ilaria Grisendi, Federica Assenza, Manuela Napoli, Claudio Moratti, Claudio Pavone, Lara Bonacini, Giovanna Di Cecco, Serena D’Aniello, Francesca Romana Pezzella, Giovanni Merlino, Fabrizio Piazza, Alessandro Pezzini, Andrea Morotti, Enrico Fainardi, Danilo Toni, Franco Valzania, Rosario Pascarella

**Affiliations:** 1Neurology Unit, Stroke Unit, Azienda Unità Sanitaria Locale-IRCCS di Reggio Emilia, Viale Risorgimento 80, 42123 Reggio Emilia, Italy; grisendi.ilaria@ausl.re.it (I.G.); assenza.federica@ausl.re.it (F.A.); valzania.franco@ausl.re.it (F.V.); 2Neuroradiology Unit, Azienda Unità Sanitaria Locale-IRCCS di Reggio Emilia, Viale Risorgimento 80, 42123 Reggio Emilia, Italy; napoli.manuela@ausl.re.it (M.N.); moratti.claudio@ausl.re.it (C.M.); pavone.claudio@ausl.re.it (C.P.); bonacini.lara@ausl.re.it (L.B.); dicecco.giovanna@ausl.re.it (G.D.C.); daniello.serena@ausl.re.it (S.D.); pascarella.rosario@ausl.re.it (R.P.); 3Stroke Unit, Department of Neuroscience, San Camillo Forlanini Hospital, 00152 Roma, Italy; frpezzella@gmail.com; 4Stroke Unit and Clinical Neurology Udine University Hospital, 33100 Udine, Italy; giovanni.merlino@asufc.sanita.fvg.it; 5CAA and AD Translational Research and Biomarkers Laboratory, School of Medicine and Surgery, University of Milano-Bicocca, Via Cadore 48, 20900 Monza, Italy; fabrizio.piazza@unimib.it; 6Department of Medicine and Surgery, University of Parma, Stroke Care Program, Department of Emergency, Parma University Hospital, 43126 Parma, Italy; alessandro.pezzini@unipr.it; 7Neurology Unit, Department of Clinical and Experimental Sciences, University of Brescia, 25121 Brescia, Italy; andrea.morotti85@gmail.com; 8Neuroradiology Unit, Department of Experimental and Clinical Biomedical Sciences, University of Florence, 50121 Florence, Italy; enrico.fainardi@unifi.it; 9Department of Human neurosciences, University of Rome La Sapienza, 00185 Rome, Italy; danilo.toni@uniroma1.it

**Keywords:** convexity subarachnoid hemorrhage, MRI, cerebral amyloid angiopathy, CAA-related inflammation, endocarditis, cerebral venous thrombosis, PRES, RCVS, intracranial stenosis

## Abstract

Spontaneous convexity subarachnoid hemorrhage (cSAH) is a vascular disease different from aneurysmal SAH in neuroimaging pattern, causes, and prognosis. Several causes might be considered in individual patients, with a limited value of the patient’s age for discriminating among these causes. Cerebral amyloid angiopathy (CAA) is the most prevalent cause in people > 60 years, but reversible cerebral vasoconstriction syndrome (RCVS) has to be considered in young people. CAA gained attention in the last years, but the most known manifestation of cSAH in this context is constituted by transient focal neurological episodes (TFNEs). CAA might have an inflammatory side (CAA-related inflammation), whose diagnosis is relevant due to the efficacy of immunosuppression in resolving essudation. Other causes are hemodynamic stenosis or occlusion in extracranial and intracranial arteries, infective endocarditis (with or without intracranial infectious aneurysms), primary central nervous system angiitis, cerebral venous thrombosis, and rarer diseases. The diagnostic work-up is fundamental for an etiological diagnosis and includes neuroimaging techniques, nuclear medicine techniques, and lumbar puncture. The correct diagnosis is the first step for choosing the most effective and appropriate treatment.

## 1. Introduction

Spontaneous non-aneurysmal convexity subarachnoid hemorrhage (cSAH) accounts for approximately 6% of all SAHs and is a different entity in comparison with aneurysmal-related SAH [1]. Both are due to spontaneous hemorrhage into the subarachnoid space, between the pia and arachnoid membranes, but cSAH is limited to the cortical surfaces, without extending into the sylvian or hemispheric fissures, basal cisterns, brain parenchyma, or ventricles. Conversely, aneurysmal SAH typically occurs around the circle of Willis within the basal cisterns, with subarachnoid bleeding and intracerebral hemorrhage (ICH) concentrated around the site of the aneurysm [2].

cSAH might have different underlying causes, and age is important to select the most probable causes. In younger people, the most probable causes include reversible cerebral vasoconstriction syndrome (RCVS) with or without posterior reversible encephalopathy syndrome (PRES), cerebral venous thrombosis (CVT), and others rarer diseases [1,3]. In people > 60 years, cerebral amyloid angiopathy (CAA) is the most common cause [1,4]. We have to consider that some diseases do not have a definite age preference. For example, CVT may affect people in all age groups with different underlying triggers (thrombophilia in young people and cancer in older people). CAA is a relevant cause of cSAH because the underlying disease has lifelong implications and not only for the individual cSAH episodes, which often recover without residual symptoms. Indeed, CAA presenting with cSAH confers an increased risk of lobar hemorrhage [by 13% per year (95% CI 10% to 17%)], recurrent cSAH [by 11% per year (95% CI 8% to 15%)] and even ischemic stroke [by 5% per year (95% CI 3% to 8%)] [5,6]. Moreover, cSAH may occur together with ICH and other CAA-related manifestations, such as CAA-related inflammation, which in turn may have a therapeutic pathway [7,8]. The deposition of amyloid β (Aβ) in cerebral amyloid angiopathy (CAA) likely stems from an imbalance between the production and clearance of Aβ, which is a degradation product of amyloid precursor protein. This imbalance leads to an accumulation of Aβ in the small leptomeningeal and cortical blood vessels, causing them to become brittle and fragile [9]. Consequently, these weakened vessels are prone to blood leakage into the subarachnoid space. The precise triggers for cerebral amyloid angiopathy-related inflammation (CAA-ri) are still unknown. However, it is suspected that an inappropriate perivascular immune response, primarily characterized by inflammatory reactivity to vascular Aβ deposition, occurs within the meningeal and cortical vessels. [10,11]. 

Currently, attention towards CAA is considerable and probably increasing, as is attention towards cSAH as a manifestation of CAA. However, this is not the only cause, and it is sometimes appropriate to remember that in individual patients, some other causes likely deserve consideration, if only to be excluded. This is because patients enrolled in the derivation and validation cohorts of the Boston criteria 2.0 [12] are highly selected by centers with a marked focus on the disease and its differential diagnosis. Therefore, the extensive application of the criteria requires a preliminary stage of differential diagnosis. Additionally, lowering the age threshold in the Boston criteria 2.0 [12] to 50 years makes the topic of the differential diagnosis of cSAH with other diseases even more delicate, especially in patients without a previous vascular history. The aim of this review is, therefore, to draw attention to the etiological differential diagnosis of cSAH beyond CAA.

The topic of the review was addressed through a dedicated literature review using public databanks (PubMed and Enbase) targeted to non-aneurysmal convexity SAH with a time filter for the publication year after 2010, limited to the papers written in the English language and exploring the references of the main papers for missing studies. The quality of evidence and reporting for some causes of cSAH was low, and this is the main reason why we selected a scoping review approach. 

## 2. Epidemiology and Clinical Presentation

Few studies are available about the incidence of cSAH in the general population. Among these, a population study based on hospital records from South Australian public hospitals [1] identified 742 cases with SAH, of which 41 (6%) cases were cSAH, with a minimum population annual incidence of 5.1 per million (95% confidence interval, 3.7–7.0). Median age was 70 years (interquartile range, 48–79). 

In patients over 60, CAA is the most common cause, often manifesting transient focal neurological episodes (TFNEs) or as transient ischemic attack (TIA)-like events [13,14]; in patients less than 60 years old, RCVS is the most common cause [2,3]. However, in several cases, cSAH was subtle and initially missed, suggesting that many cases of cSAH may remain undiagnosed [1]. Some cases (especially of CAA-related TFNS) may not be investigated or not seek medical attention at all. Therefore, true cSAH incidence is probably at least double the available estimate.

cSAH presents with a range of symptoms that differ based on the cause and the patient’s age. Common symptoms include headaches, impaired consciousness, seizures, confusion, transient focal neurological episodes, persistent focal neurological deficits, visual abnormalities, nausea, and vomiting. However, cSAH has rarely been categorized as a type of non-aneurysmal subarachnoid bleeding, likely due to its diverse presentations and etiologies. Therefore, both epidemiology and clinical presentation data are incomplete. Contrary to Kumar et al. [3], Beitzke et al. [2] found cSAH to be significantly more prevalent in patients aged 60 and above compared to younger individuals. This discrepancy can be attributed to the atypical symptoms of subarachnoid bleeding often seen in older patients, because nearly half of the patients aged ≥ 60 years (45%) had TFNEs, and 58% did not experience headaches [2]. Consequently, the association with cSAH might be easily overlooked or misinterpreted as a secondary phenomenon. It is crucial to note that CT scans may miss cSAH if not performed promptly after a presumed TIA, since blood can become isodense within a few days of the bleeding. 

## 3. Main Causes and Examples

### 3.1. CAA

CAA is one of the main causes of spontaneous cSAH, and the Boston 2.0 criteria [15] added cSAH as a hemorrhagic manifestation of the disease suitable for diagnosis. One of the main clinical manifestations of cSAH is represented by TFNEs or amyloid spells [13]. 

TFNEs are recurrent, stereotyped, and spreading focal neurological symptoms or signs, typically lasting several minutes. Recognizing TFNEs is clinically important because they can be a key diagnostic indicator of CAA beyond intracerebral hemorrhage (ICH) and may precede symptomatic ICH. Avoiding the use of antithrombotics after a misdiagnosis as a TIA can reduce the risk of ICH. Although the pathogenesis of amyloid spells is likely heterogeneous—ranging from the narrowing of amyloid-laden small vessels to seizure-like mechanisms related to microbleeding or focal convexity subarachnoid hemorrhages—they may serve as a useful clinical marker for CAA. The main pathophysiological hypothesis underlying TFNEs is cortical spreading depression [14]. In CAA patients presenting with TFNE, cortical superficial siderosis (cSS) or cSAH are more frequent than in those without TFNE (50% vs. 19%; *p* = 0.001). In addition, 50% of TFNE patients had symptomatic lobar ICH over a median period of 14 months. The meta-analysis showed a risk of symptomatic ICH after TFNE of 24.5% (95% CI 15.8–36.9%) at 8 weeks, related neither to clinical features nor to previous symptomatic ICH [13]. In a series of patients with siderosis and/or cSAH suspected of having CAA, follow-up data were available for 76 out of 90 patients (84%). Among these, ten patients with cSS and/or cSAH (29%, median time from scan: 21 months) experienced a symptomatic cerebral bleeding event (either ICH or cSAH) during follow-up (average time to event: 34 months). In contrast, only one event (2.4%, 25 months from scan) occurred in patients without SS/cSAH (time to ICH: 25 months) (*p* = 0.001) [16]. 

The prognosis for CAA-related cortical subarachnoid hemorrhage (cSAH) has been detailed in a recent study [5], revealing the following risks per patient-year: intracerebral hemorrhage (ICH) at 13.2% (95% CI 9.9–17.4), recurrent cSAH at 11.1% (95% CI 7.9–15.2), combined ICH or cSAH at 21.4% (95% CI 16.7–26.9), ischemic stroke at 5.1% (95% CI 3.1–8), and mortality at 8.3% (95% CI 5.6–11.8). In multivariable analyses, patients with probable CAA had a significantly higher risk of ICH (HR 8.45, 95% CI 1.13–75.5, *p* = 0.02) and cSAH (HR 3.66, 95% CI 0.84–15.9, *p* = 0.08) compared to those with possible CAA. However, there was no significant difference in the risk of ischemic stroke (HR 0.56, 95% CI 0.17–1.82, *p* = 0.33) or mortality (HR 0.54, 95% CI 0.16–1.78, *p* = 0.31). CAA-related inflammation or ARIA (amyloid-related imaging abnormalities)-like events are another CAA-related manifestation potentially related with both TFNEs and cSAH, sometimes with a leptomeningeal involvement and with different diagnostic criteria. Clinical and imaging criteria have been described for the diagnosis of probable CAA-related inflammation, demonstrating high sensitivity (82%) and specificity (97%) [17]. Diagnostic criteria rate only classical parenchymal and mixed forms. Criteria of ARIA-E (essudation) include sulcal, parenchymal, and mixed sulcal and parenchymal forms both supratentorial and infratentorial [18,19]. In addition, CAA-related inflammation is a spontaneous model for iatrogenic ARIA, useful for identifying the pathophysiology, and its autoimmune-mediated pathway is supported by the identification of anti-amyloid beta antibodies in CSF [8,20,21]. The relevance of diagnosis is strongly evident both for avoiding harmful treatments and for starting immunosuppressive therapy in CAA-related inflammation [7,8,21,22]. Figure 1 illustrates an example of cSAH related to CAA. 

### 3.2. Hemodynamic Arterial Stenosis

Extracranial or intracranial artery stenosis, including internal carotid artery (ICA) or middle cerebral artery (MCA) stenosis, is rarely documented in cases of cSAH [23]. Additionally, the occurrence of an acute ischemic infarct alongside cSAH has been infrequently reported, typically being ipsilateral to the infarct [3]. 

The underlying pathophysiology that correlates acute ischemic infarct with cSAH is not completely understood. The development and evolution of collateral circulation likely play a pivotal role in this condition. In cases of atherosclerotic disease, progressive arterial occlusion can lead to the formation of extensive collateral circulation. While this collateral circulation serves as a compensatory protective mechanism against cerebral ischemia, the pathophysiology and clinical outcomes associated with concurrent acute ischemic infarct and cSAH may involve more intricate mechanisms.

Cho et al. [24] observed two abnormal angiographic patterns. The first one is characterized by the robust development of leptomeningeal collaterals, which shifted the ACA-MCA border zone towards the MCA side compared to the unaffected side, along with reduced MCA vessel size but maintained flow. The second pattern involves dilatation, associated with inadequate development of leptomeningeal collaterals, resulting in cerebral vessel dilation and suggesting impaired autoregulation. The authors concluded that patients exhibiting the dilatation pattern tended to experience more strokes and border zone infarcts than those with the shift pattern, indicating that dilatation might pose an angiographic risk factor for symptomatic watershed ischemic stroke. Furthermore, arteriogenesis involves the formation of new vessels that are more susceptible to injury due to altered blood flow dynamics, shear stress, inflammation, endothelial activation, and remodeling. This vulnerability can predispose these vessels to bleeding, akin to what is observed in moyamoya disease. [25].

Progressive ICA occlusion typically facilitates the development of collateral circulation. When the compensatory flow from the circle of Willis is insufficient, alternative pathways are activated through extracranial-to-intracranial anastomoses and leptomeningeal collaterals. These pial arteries connect the cortical branches of the MCA, anterior cerebral artery (ACA), and posterior cerebral artery (PCA). A weakening of the arterial wall may contribute to cortical bleeding, as indicated in a previous report on cSAH that identified dilated pial collaterals via angiography. Notably, prior cases of cSAH have consistently occurred on the symptomatic side, primarily within the MCA-PCA watershed area [26]. In the setting of high-grade chronic ICA stenosis, especially when there is no collateral support via the circle of Willis, compensation flow is provided via the PCA. If an acute increase in cerebral blood flow is required, the hyperflow through the PCA may favor the rupture of dilated, fragile pial PCA-MCA leptomeningeal collaterals [27,28]. These dilated collaterals reflect exhausted vasoreactivity and could, therefore, increase the risk of cerebral hyperperfusion syndrome after carotid revascularization.

The etiology of acute ischemic stroke occurring alongside cSAH varies based on location and underlying pathophysiology. Although this presentation is uncommon, acute ischemic stroke is most frequently associated with severe ICA stenosis. When cSAH accompanies acute ischemic stroke, it typically occurs ipsilaterally, while contralateral cases are rarely reported. Ipsilateral cSAH is attributed to the vulnerability of compensatory collaterals, which can rupture and bleed at any moment [28]. Contralateral cSAH is much rarer and may have a slightly different underlying pathophysiology [29]. Chronic internal carotid artery (ICA) stenosis triggers various compensatory mechanisms aimed at preserving cerebral blood flow. One significant adaptation involves contralateral vasculature dilation to maintain adequate perfusion. However, if these vessels dilate excessively beyond physiological limits, it can compromise their structural integrity, increasing the risk of vessel fragility and rupture [29]. Another compensatory mechanism in response to chronic ICA stenosis involves the establishment of contralateral collateral circulation. This adaptation aims to maintain perfusion to brain regions that would otherwise be supplied by the occluded vessels. However, similar to dilated vessels, collateral vessels are typically fragile, increasing their susceptibility to rupture. This can lead to hemorrhage, often presenting as cSAH [29]. A study involving 384 patients with symptomatic ischemic stroke aimed to measure the incidence of cSAH with acute ischemic stroke [30]. 

Out of 384 patients, 6 experienced arterial acute ischemic strokes, while 2 had venous acute ischemic strokes. Notably, only two patients (0.5%) were diagnosed with cSAH within 4.5 h of presentation, with another two cases identified within six days. A separate study investigating the incidence of cSAH alongside acute ischemic strokes found that over half of the patients with cSAH had a major artery occlusion. These results indicate that significant arterial obstruction, particularly involving the extracranial ICA, is a primary cause of cSAH. 

Carotid revascularization is a well-established procedure for managing high-grade ipsilateral symptomatic carotid disease, demonstrating a marked reduction in annual stroke rates and improved patient outcomes [23]. Consequently, similar treatments and medical therapies may be advantageous for patients with concurrent cSAH. Treatment can be tailored to individual patient needs but typically includes antiplatelet agents (such as aspirin or clopidogrel), statins, antihypertensives, and lifestyle modifications. The presence of cSAH, even without a stroke, may indicate altered hemodynamics and could warrant consideration for carotid revascularization [31].

An example of this mechanism is provided in Figure 2.

### 3.3. Infective Endocarditis

cSAH is a rare finding in infective endocarditis (IE), occurring in about 5% of cases. Intracranial infectious aneurysms (IIAs) are found in 2–8.9% of IE cases (Figure 3). A recent study of Boukobza et al. [32] with a retrospective design on 240 IE patients identified a 12.9% prevalence of cortical cSAH in IE patients. From the clinical point of view, the most significant difference observed was the occurrence of headaches in the cSAH group: 16.1% compared to 1.4% in the non-cSAH group (3.3% in the overall population), with no reports of thunderclap headaches in the cSAH group. Confusion was the sole symptom in three patients, and acute meningeal syndrome was observed in one patient with IIA. These findings align with previous small studies [33], suggesting that cSAH is often a silent event in IE, potentially leading to an underestimation of its prevalence. Unruptured IIAs are sometimes incidental findings (8–34%), with non-invasive imaging techniques improving their detection [34]. An old study [35] on 489 IE patients between 1980 and 1996 found 8 patients with SAH, with 5 cases limited to the frontal and parietal sulci. However, it should be kept in mind that the concepts of cSAH and FLAIR imaging was not yet established. FLAIR imaging can precisely identify subtle cSAH [36] though artifacts from high oxygen concentrations in intensive care patients or post-contrast MRI, but some conditions, like meningitis, can mimic SAH [37]. However, in other diseases (e.g., CAA and CAA-related inflammation), the conclusion of cSAH on the basis of the FLAIR sulcal hyperintensity data may be misleading, as there are other causes of proteinaceous leakage at the leptomeningeal level. In a recent study, [33] identified 9 IE cases out of 88 patients with non-traumatic cSAH, with unilateral cSAH in nearly half and a concurrent acute ischemia in 44.4% of cases. Most cases of cSAH were unilateral (64.7%) and associated with cerebral microbleeds (CMBs) (76.5%) and acute ischemia (70.6%) [33]. Two recent isolated cases reported cSAH concurrent with IE-IIAs, located near [38] and remotely [39] from the IIA. The diagnostic impact of sulcal blood susceptibility artifact thickness in IIA cases is notable. Unlike Kumar et al. [3], Boukobza et al. [32] found enhancement at the cSAH site on post-contrast T1WI in 16.1% of cases. Malhotra et al. [40] suggested that cSAH and cSS are not closely linked with IIAs in IE. The distribution of cSS correlated with initial bleeding location and amount, similar to findings after acute aneurysmal and traumatic SAH [41,42]. There are data in an opposite direction, i.e., a lower cSS prevalence [32], possibly due to the small blood volume in IE-cSAH. This pattern aligns with recurrent occult subarachnoid bleeding as a cause of cSS development [16,42,43,44]. In the cohort described by Boukobza et al. [32], valvular vegetations and their length were significantly associated with cSAH (*p* < 0.0001), but not with other lesions (perivalvular abscess, brain macro-hemorrhages, visceral emboli, etc.). 

IE abscesses at the cerebritis stage rarely cause cSAH, but IE should be considered when abscesses or meningitis coexist with CMBs [45]. No cSAH recurrence or neurologic events indicating new hemorrhage were observed, and no second angiograms or CTA were performed during follow-up. Patient outcomes were generally favorable, suggesting that cSAH unrelated to IIA is not a marker of poor prognosis.

Cardiac findings (high prevalence of valve vegetations, mitral valve lesions, and vegetation length ≥ 15 mm) support hypotheses about cSAH mechanisms in IE, involving septic embolic material from valve vegetations, capillary rupture, or focal arteritis [46]. cSAH near IIA, often with more blood evident in MRI, might result from IIA rupture or fissuration. While cSAH is rare in non-infectious IAs, it is relatively common in IIA, likely due to their distal locations.

### 3.4. Cerebral Venous Thrombosis

CVT has long been recognized as a cause of parenchymal hemorrhage. cSAH, either isolated or in combination with parenchymal hemorrhages, represents a rare manifestation of CVT, accounting for 6.5% of patients in the largest series [47]. The literature mostly consists of isolated case reports or small series documenting this occurrence. Out of 41 cases within 2016 with comprehensive radiological data on CVT-related SAH, 32 presented with isolated SAH and 26 underwent MRI evaluations [48,49,50,51,52,53,54,55,56,57,58,59,60,61,62,63,64,65]. Notably, there is a lack of systematic studies investigating SAH in the context of CVT using T2* sequences, thus leaving the incidence of this association unclear. A single study [47] aimed to assess the occurrence and radiological features, particularly using T2*, of SAH linked with CVT, focusing on the presence, localization, and distribution of associated cortical venous thrombosis (CoCVT) in 22 patients, all of whom exhibited CVT presenting as SAH without concomitant parenchymal hemorrhage. Both non-contrast computed tomography (NCCT) (within 24 h of admission) and MRI were used for diagnosing cSAH in the selected cohort, being cSAH found on NCCT in 7/22 (31.8%) cases, unilaterally and limited to few sulci, and on FLAIR sequence in 21/22 (95.5%) cases, unilaterally and limited to a few sulci at the convexities. Interestingly, in 15 patients, the diagnosis of SAH was missed on CT scans due to the small amount of blood present. The diagnosis was instead based on MRI, which is well-established as being superior to CT for the presumptive diagnosis and localization of acute and subacute low-grade SAH [66,67,68]. The typical MRI findings often reveal hyperintensities in the subarachnoid spaces on FLAIR sequences, occasionally accompanied by hypointensities at the same locations on T2* sequences. Thrombotic involvement of the dural sinuses was identified in 21 out of 22 cases (95.5%), with both the superior sagittal sinus (SSS) and lateral sinus (LS) affected in 12 out of 22 patients (54.5%). All cases exhibited cortical vein thrombosis (CoVT), characterized on imaging by tubular, serpiginous structures with strong hypointensity on T2* imaging and corresponding hyperintensity on T1 imaging in five cases. FLAIR and DWI sequences showed hyperintensity in only one case. In all instances, CoVT was adjacent to the area of SAH, and cortical thrombus was consistently continuous with sinus thrombosis when present. The vein of Labbé was the most frequently involved (14 out of 21 patients), always associated with ipsilateral LS thrombosis. Frontal vein involvement occurred in 11 out of 21 patients, with 10 of these cases also showing SSS thrombosis. Multiple associations with CoVT were common. Bilateral frontal vein thrombosis was observed in two cases, correlating with bilateral SAH in the frontal regions. Limited SAH in multiple areas was noted in two other cases. Isolated CoVT was seen in one instance, affecting the frontal and parietal veins on the same side, with SAH localized to the ipsilateral fronto-parietal convexity.

CVT is a recognized but uncommon cause of cSAH. Diagnosing CVT in patients with SAH is essential, as it necessitates anticoagulation therapy. While identifying CVT is relatively straightforward in cases of isolated SAH with evident sinus thrombosis, it becomes significantly more challenging when only cortical vein thrombosis is present. Firstly, diagnosing cortical vein thrombosis can be difficult even in the absence of SAH, as the number and location of cortical veins can vary widely, with only the largest veins visible on MRV or CT venography. Secondly, when SAH is confined to the convexity, differentiating it from cortical vein thrombosis on FLAIR sequences can be nearly impossible. In these instances, T2* sequences become critical, as CoVT is visualized as a hypointense tubular structure, whereas SAH appears as subtle hemosiderin deposits. [66,67,68].

The location of cSAH can significantly aid in diagnosis. For instance, cSAH in the fronto-parietal regions, especially if bilateral or located in the parasagittal sulci, should raise suspicions of thrombosis of the SSS and/or the frontal and parietal veins. In contrast, if cSAH is restricted to the posterior temporal or temporo-occipital convexity, or the distal sylvian fissure, it is important to investigate for potential thrombosis of the transverse and sigmoid sinus and the vein of Labbé [47].

Several hypotheses explain the occurrence of SAH in patients with CVT. The first one is that sinus thrombosis increases venous pressure, causing the dilation and rupture of adjacent fragile cortical veins, leading to localized hemorrhages in the subarachnoid space [48]. The second one suggests secondary rupture of parenchymal hemorrhagic lesions into the subarachnoid space. The third hypothesis proposes that sinus thrombosis extending into cortical or cerebellar veins causes localized venous hyperpressure, leading to dilation and rupture of these veins, as documented in experimental studies [69,70,71]. Figure 4 shows an example of CVT-related cSAH.

### 3.5. Primary Central Nervous System Angiitis

Primary angiitis of the central nervous system (PACNS) is a rare cerebrovascular disease characterized by transmural inflammation of leptomeningeal, cerebral, and spinal vessels. The diagnostic criteria for PACNS were first proposed by Calabrese and Mallek [72] in 1988 and updated by Birnbaum and Hellmann [73] in 2009. PACNS can present with a variety of clinical symptoms and neuroimaging patterns, which are often non-specific and inconsistently reported in the literature, as noted by the recent European Stroke Organization guidelines [74]. One manifestation of PACNS is intracranial bleeding, both spontaneous ICH and cSAH, but, unfortunately, the available data are fragmentary and incomplete [75]. In particular, the description of neuroimaging data often do not differentiate between ICH and SAH, and it is impossible to retrieve the association between cSAH in the context of ICH or ischemic stroke and there are no data about the relationship between small-vessel or large-vessel PACNS and hemorrhagic findings. In the literature review carried out in the ESO guidelines [74], a hemorrhagic presentation of PACNS, including both ICH and SAH patterns, was reported in 13.6% of the patients, combining old and new ICH and micro- and macro-hemorrhages. It follows that it is not possible to describe in detail the characteristics of cSAH as a neuroradiological manifestation in the context of PACNS.

### 3.6. PRES/RCVS

Posterior reversible encephalopathy syndrome (PRES) is a clinical and neuroradiological condition associating different neurological manifestations (e.g., encephalopathy, seizures) with a typical brain vasogenic edema. After its first description almost 30 years ago [76], PRES now encompasses several radiological patterns, and a plethora of triggers and etiologies (e.g., hypertensive encephalopathy, eclampsia, autoimmune diseases, and anticancer drugs) have been identified over the years [77,78,79,80].

PRES is part of a broader category known as “reversible posterior leukoencephalopathy syndrome” (RPLS). This clinical radiographic syndrome encompasses various underlying causes that exhibit similar neuroimaging findings. It is commonly referred to by several names, including PRES, reversible posterior cerebral edema syndrome, posterior leukoencephalopathy syndrome, hyperperfusion encephalopathy, and brain capillary leak syndrome. However, none of these terms fully capture the syndrome’s nature, as it is not always reversible and often affects areas beyond just the white matter or posterior regions of the brain.

The pathophysiology of PRES remains controversial. and two main theories have been proposed [81]: (1) intense cerebral autoregulatory vasoconstriction in response to acute hypertension leading to decreased CBF, ischemia, and subsequent edema involving mainly the border zone arterial regions (support from an animal model: hypertensive rat’s pial vessels); (2) forced vasodilatation of cerebral vessels (autoregulation breakthrough) rather than vasoconstriction as the major component of PRES, resulting in extravasation of fluid into the interstitium, termed vasogenic edema (endothelial dysfunction). 

The most characteristic imaging pattern [82,83,84] is the presence of edema involving the posterior white matter of both cerebral hemispheres, especially the parieto-occipital regions, in a relatively symmetric pattern. The calcarine and paramedian occipital-lobe structures are usually spared, which helps differentiate it from PCA infarction.

Other regions of the brain are also frequently affected, such as the following:-Parietal or occipital regions: 98%.-Frontal lobes: 68%.-Inferior temporal lobes: 40%.-Cerebellar hemispheres: 30%.-Basal ganglia: 14%.-Brainstem: 13%.-Deep white matter, including the splenium: 18%.

In milder cases, there is a greater involvement of gray matter than white matter. As the extent of the edema increases, lesion confluence may develop. In addition to the typical PRES neuroimaging patterns, there are some well-described atypical patterns. 

In a recent systematic review on cerebrovascular manifestations of PRES and RCVS [85], including 1385 PRES patients, the rate of ischemic stroke was 11.2% (95% CI 7.9–15%), the rate of intracranial hemorrhage was 16.1% (95% CI 12.3–20.3%), the rate of SAH was 20.3% 7% (95% CI 4.7–9.9%), and the rate of ICH was 9.7% (95% CI 5.4–15%). However, ischemic stroke as first manifestation of PRES is underreported [86].

cSAH is not uncommon in PRES, and the putative mechanism derives from the phenomenon of breakthrough perfusion, i.e., the rupture of pial vessels in the face of severe hypertension or vasoconstriction coupled with blood pressure instability predisposing a patient to reperfusion injury. Hemorrhage type and frequency do not appear to be linked to blood pressure in patients with PRES. Microhemorrhage occurrence does not correlate with the severity or extent of edema, the presence of DWI-positive findings or the presence of enhancement [87]. At present, there are no diagnostic criteria for PRES, and both clinical and neuroimaging findings are often not specific [81]. The proposed diagnostic clues are: (1) presentation with acute clinical symptoms; (2) presence of a known risk factor for PRES; (3) reversibility of clinical and/or radiological findings; (4) ruling out of other possible causes of encephalopathy or vasogenic edema; (5) distributions of FLAIR hyperintensities compatible with typical PRES imaging patterns, and (6) vasogenic edema as demonstrated by DWI and ADC. 

Reversible cerebral vasoconstriction syndrome (RCVS) encompasses a group of conditions characterized by reversible multifocal narrowing of the cerebral arteries. Patients typically present with thunderclap headaches and may also experience neurological deficits due to brain edema, stroke, or seizures. While the prognosis is generally favorable, severe strokes can lead to significant disability or even death in some cases. RCVS has been linked to various factors, including pregnancy, migraines, the use of vasoconstrictive medications, neurosurgical procedures, hypercalcemia, unruptured saccular aneurysms, cervical artery dissection, and CVT. Notably, the individual risk factors and triggers for RCVS often appear to be unrelated and may reflect investigator biases in identifying risks. There is ongoing debate regarding the pathophysiology of PRES and RCVS, particularly whether they represent distinct entities or exist along a pathological continuum. However, both conditions are primarily associated with the reversible dysregulation of cerebral vascular function. RCVS and PRES can co-occur in some situations, such as hypertension, preeclampsia or eclampsia, autoimmune disorders, intracranial hypotension, and the use of vasoactive or cytotoxic agents. Imaging findings may overlap, as typical PRES characteristics are found in 8–38% of RCVS patients; some individuals with PRES may also exhibit reversible segmental vasoconstriction of intracranial arteries. In many cases of RCVS, thunderclap headache is the sole symptom, while others may develop focal deficits resulting from ischemic stroke, intracerebral hemorrhage (ICH), or reversible cerebral edema. The incidence of focal neurological deficits varies widely, ranging from 9% to 63%, with brain lesions including infarction (39%), brain edema (38%), convexity subarachnoid hemorrhage (33%), and lobar hemorrhage (20%) [88]. 

Brain imaging often appears normal in the early stages of RCVS. Common findings include vasogenic edema and/or FLAIR hyperintensities in the sulci, known as the “dot sign” on MRI. When infarcts occur, they are typically symmetric and located along the border zones of arterial territories. Some RCVS cases may also present with intraparenchymal hemorrhage or non-aneurysmal cSAH. The defining feature of RCVS on cerebral angiography is multifocal segmental vasoconstriction of the cerebral arteries.

A recent systematic review analyzing data from 2746 RCVS patients reported an ischemic stroke rate of 15.9% (95% CI 9.6–23.4%), an intracranial hemorrhage rate of 22.1% (95% CI 10–39.6%), an SAH rate of 20.3% (95% CI 11.2–31.2%), and an ICH rate of 6.7% (95% CI 3.6–10.7%). According to a neuroradiological review [89], cSAH accounts for 38% of all hemorrhagic complications associated with RCVS. It is typically mild and found in the cerebral sulci near the vertex. In nearly 38% of cases, it is bilateral, affecting one cortical sulcus in 36% of patients, two sulci in 26%, and more than three sulci in 38%. Blood rarely extends to the sylvian fissures or the ambient cisterns. The most common locations include the frontal lobe (79% of cases) and the parietal region (31%), while the occipital and temporal lobes are infrequently involved (23% and 10%, respectively). In about half of the cases, SAH is associated with either intracerebral hemorrhage or ischemic stroke. Abnormalities in cerebral angiography are the key diagnostic indicator of RCVS. These angiographic changes are dynamic, evolving over time from distal to proximal vessels, and are characterized by intermittent focal narrowing that resembles a “sausage on a string” appearance in the circle of Willis and its branches. Accurate diagnosis of RCVS can often be achieved through a thorough history and the results of initial brain and non-invasive vascular imaging [89,90,91].

### 3.7. Vascular Malformations

Vascular malformations (dural artero-venous fistulae or DAVF and artero-venous malformations or AVM) are only very rarely the cause of isolated cSAH without concomitant aneurysms. In particular, the pattern of SAH associated to DAVF or AVM in cranial or spinal location is often diffuse and involves the posterior cranial fossa [92]. When a vascular malformation is suspected, the diagnostic pathway includes a digital subtraction angiography.

## 4. Diagnostic Pathway

The main aim of the initial diagnostic pathway of cSAH is to distinguish it from aneurysmal SAH [93]. After that, identifying the cause of cSAH requires a complex diagnostic pathway and it can be challenging. As with other diseases, clinical history and examination might help to formulate diagnostic hypotheses and to choose the investigations. Neuroradiological techniques are the cornerstone of etiological investigations of cSAH, starting from the brain NCCT scan. Small amounts of subarachnoid blood can be easily overlooked, and the coronal view is particularly useful for identifying subtle bleeds. CT scanning has a high sensitivity (98%) for detecting SAH within 6 h of onset, though this drops to around 90% after 6 h [94]. In cSAH, NCCT likely has a lower sensitivity due to the smaller volume of blood involved. In RCVS, a CT angiography (CTA) may show segmental narrowing of multiple intracranial arteries, indicating vasospasm in various vascular territories, usually after 14 days from symptoms onset. A CT venography is appropriate if CVT is suspected. A summary of the diagnostic clues for cSAH is illustrated in Figure 5. 

In the acute phase of cSAH, an MRI of the brain often reveals elevated T2/FLAIR and T1 signals within the affected sulcus. However, shortly after presentation, the scan may appear normal if there is only a minor leptomeningeal protein leak [95]. Susceptibility-weighted imaging (SWI) and gradient-recalled echo (GRE) are T2* MRI sequences that identify paramagnetic materials as hypointense signals. In the early stages of cSAH, this susceptibility effect may not be evident. However, as time passes and blood products undergo degradation, resulting in hemosiderin deposits, the SWI and GRE signals become hypointense, resembling superficial siderosis. By this stage, hyperintense signals on T2/FLAIR typically diminish. Most insights into the timing of MRI signal changes in subarachnoid hemorrhage come from studies of aneurysmal SAH. It is estimated that the sensitivity and specificity of FLAIR for detecting SAH within the first two days after onset are 100% [96]. In addition, FLAIR had a greater sensitivity than CT and GRE imaging within 4 days (sensitivities: 100% for FLAIR, 71.8% for CT, 37.5% for GRE) and 4–15 days (sensitivities: 100% for FLAIR, 50% for CT, 30% for GRE) after suspected onset of low-grade SAH [97]. GRE and FLAIR were each more sensitive than CT when MRI is performed >4 days after symptom onset (sensitivities: 100% for GRE, 87% for FLAIR, 75% for CT) [95]. Combining SWI and FLAIR resulted in a superior rate of SAH detection compared to CT when MRI was performed within 6 days of symptom onset [36]. Blood secondary to SAH was better identified on GRE than on FLAIR 3 months after SAH (sensitivities: 2% for FLAIR, 35% for GRE; specificities: 98% for FLAIR, 87% for GRE) [98]. In some reports deriving from aneurismal SAH, GRE, if performed within 90 days of SAH onset, might allow doctors to estimate the approximate day on which the SAH occurred [99]. However, GRE and SWI cannot identify an SAH if imaging is performed too early to allow for hemoglobin decomposition to deoxyhemoglobin [36]. In addition, the double inversion recovery (DIR) sequence, which employs two inversion recovery pulses to suppress white matter and cerebrospinal fluid, has a higher sensitivity than CT, 2D FLAIR, 3D FLAIR, GRE, and SWI sequences for diagnosing SAH 14–16 days after symptom onset (sensitivity and specificity inferred to be 100%) [100]. An old study [95] addressing the sensitivity and specificity of five MRI sequences in acute and subacute SAH revealed that the sensitivity to SAH across the five MR sequences examined varied significantly, ranging from 50% to 94% in acute SAH cases and from 33% to 100% in subacute SAH cases. The sequences that demonstrated the highest sensitivity were FLAIR and T2*, with T2* showing slightly better performance than FLAIR. Specifically, T2* had a sensitivity of 94% within the first 4 days after the ictus and 100% between 4 and 14 days. The overall specificity exceeded 98.5% (95% CI: 96.75–100%).

Dynamic, transient MRI changes in sulcal T2/T1 and SWI signals can also be observed in individuals with Alzheimer’s disease undergoing monoclonal anti-amyloid therapy. These changes are referred to as amyloid-related imaging abnormalities with hemosiderin deposits (ARIA H). Additionally, similar transient sulcal alterations can occur without evidence of superficial siderosis or hemorrhage, likely due to protein-rich fluid leakage into the leptomeningeal space, known as ARIA E [101]. A spontaneous ARIA-like pattern [8] has been described in patients with CAA, and it also involves a leptomeningeal compartment. MRI and MR angiography (MRA) can provide important clues to the underlying cause of cSAH, according to the updated Boston criteria’s [12] increased diagnostic sensitivity and included non-hemorrhagic presentation. Moreover, cSAH [102] is included in the hemorrhagic presentations of CAA in Boston criteria 2.0, and TFNEs are included in the clinical presentations to which criteria can be applied. 

In PRES, vasogenic edema primarily occurs in the posterior parietal and occipital areas, but it can also affect the brainstem and frontal lobes in some cases. This edema usually presents as a hyperintense signal on FLAIR images and shows increased diffusion on apparent diffusion coefficient (ADC) maps, often resembling a “finger glove” pattern as it extends along the white matter pathways [83]. These alterations typically arise in the subcortical white matter but can also involve the gray matter. Patients with CAA-related inflammation may show single or multiple confluent hyperintense lesions on FLAIR images, indicative of vasogenic edema. This is often accompanied by varying degrees of increased leptomeningeal enhancement on post-contrast T1 and FLAIR scans [17].

In RCVS, CTA or MRA show diffuse and segmental narrowing or “beading” of multiple vessels. The improved resolution of these scans has reduced the need for digital subtraction angiography. 

In CAA-related cSAH, Pittsburg compound B and other amyloid-specific ligands used in PET imaging show increased uptake in the cortical region [103].

A lumbar puncture is typically advised for patients experiencing “thunderclap” headaches when a CT scan does not reveal signs of SAH and there are no contraindications. The primary objective is to assess the presence of red blood cells and xanthochromia. While red blood cells might result from a traumatic tap, xanthochromia is a more specific indicator of hemorrhage, although its sensitivity varies depending on the timing of the lumbar puncture relative to headache onset. Xanthochromia can appear as early as 2 h after the onset of symptoms, with peak sensitivity occurring around 12 h later. If imaging confirms cSAH, a lumbar puncture is generally unnecessary unless there are concerns about infection or other immune-inflammatory conditions. Most cerebrospinal fluid (CSF) analyses return normal results but may show non-specific changes, such as elevated protein levels and pleocytosis. Additionally, CSF testing can aid in diagnosing CAA and CAA-related inflammation, as it is supported by growing evidence [104,105]. 

## 5. Conclusions

The differential diagnosis of cSAH includes several different diseases beyond CAA. Individual cases should be managed considering the full range of differential diagnoses, because the subsequent treatment is different and in some cases opposite, as for example abstention from antithrombotic medications in CAA versus antimicrobial drugs in infective endocarditis versus anticoagulation in CVT.

## Figures and Tables

**Figure 1 jcm-13-04382-f001:**
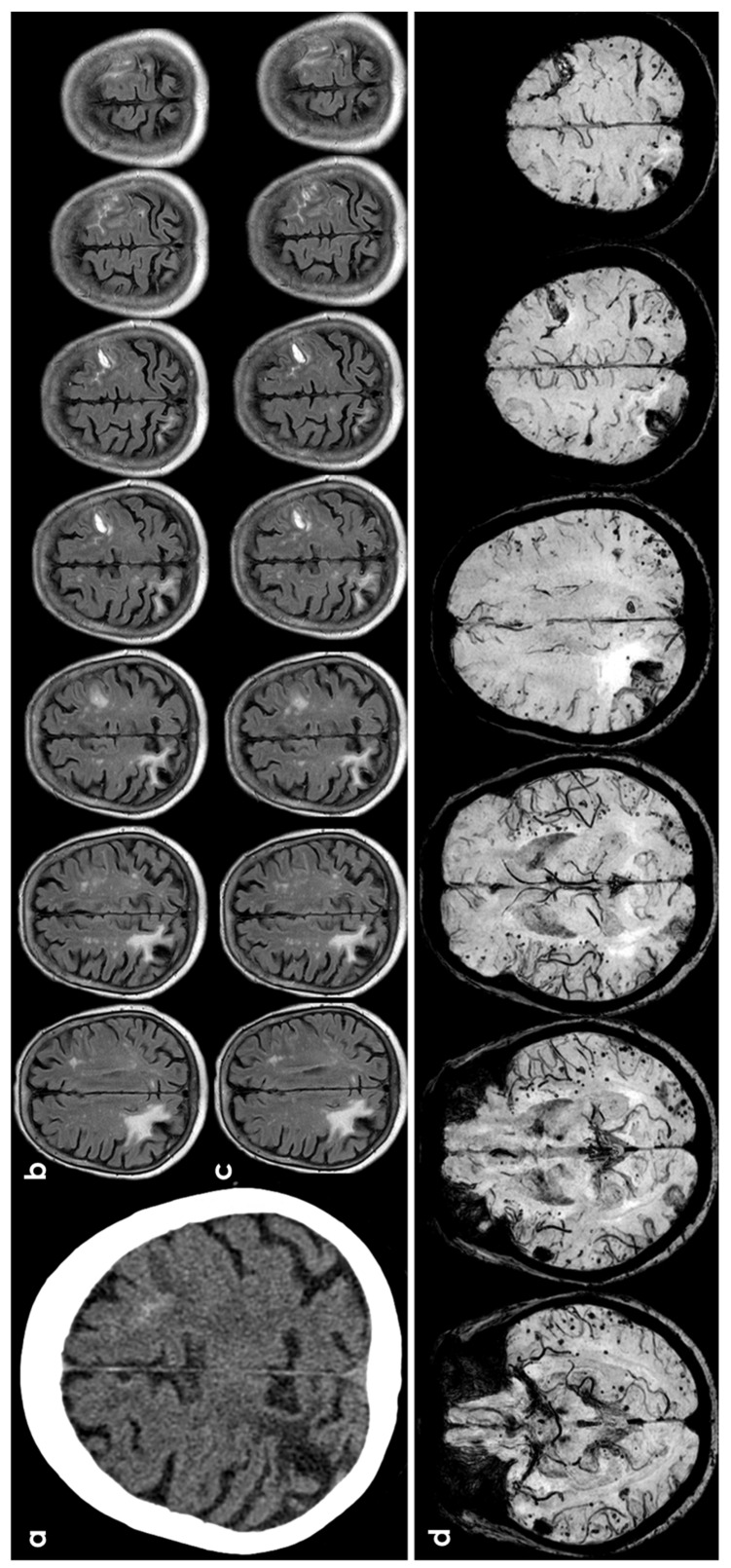
cSAH in a patient with probable CAA according with Boston criteria 2.0 [12]. Panel (**a**) shows a non-contrast computed tomography (NCCT) axial slice with the hyperdense content in one left anterior frontal sulcus, suggesting cSAH. Panels (**b**,**c**) show sequential axial FLAIR slices before (**b**) and after GBCA. Spontaneous contrast enhancement in more than one left anterior frontal sulcus is accompanied by cortical effacement and subcortical white matter hyperintensity in the same site. Panel (**d**) shows an SWI sequence reconstructed in axial plane with minimum, finding several lobar microbleeds, at least two macrobleeding foci and multifocal cortical superficial siderosis.

**Figure 2 jcm-13-04382-f002:**
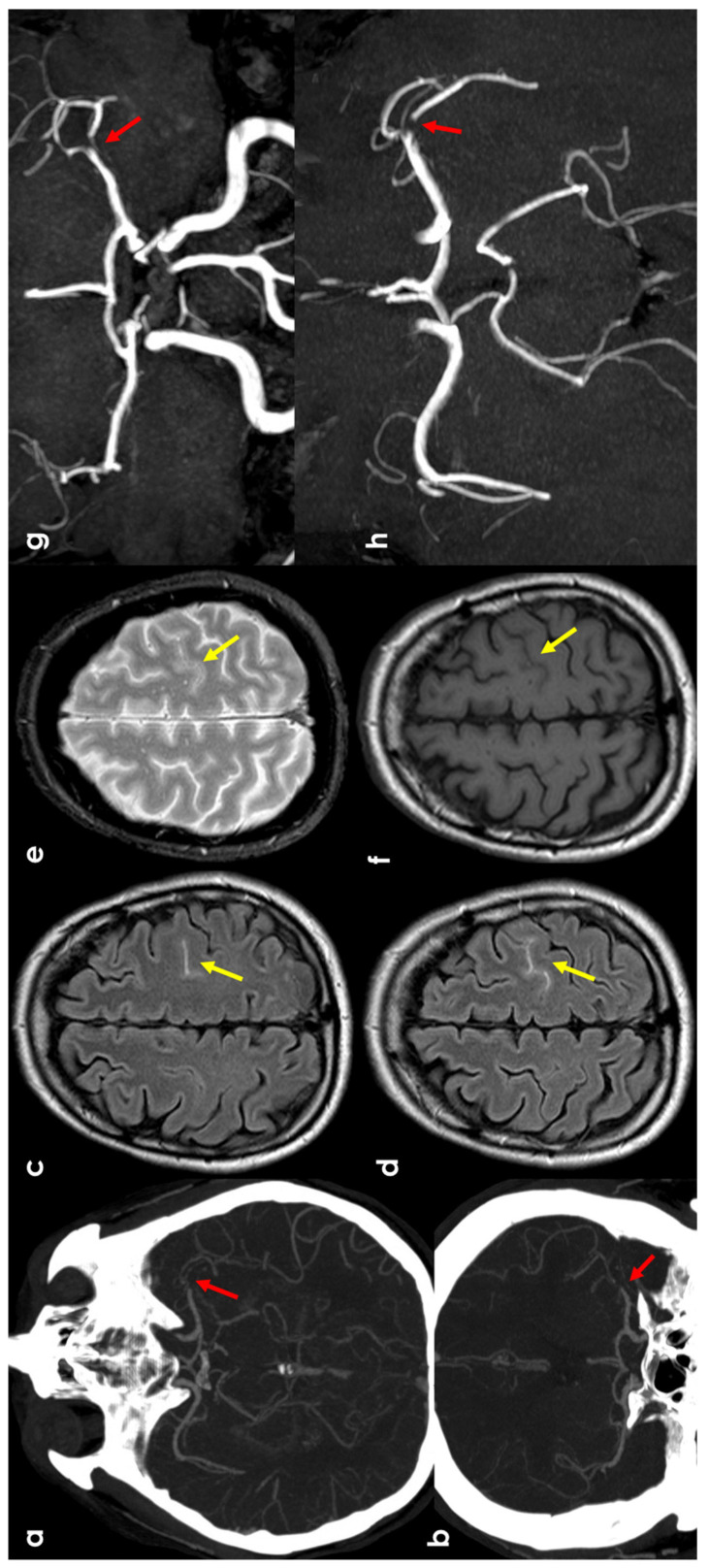
cSAH in a patient with left distal M1 MCA acute steno-occlusion. Panels (**a**,**b**) show CT angiography reconstructed with the maximum intensity projection/MPR tool in the axial and coronal planes, respectively, showing the focal lack of left MCA opacification corresponding to vessel occlusion (red arrows). Panels (**c**,**d**) show axial FLAIR-MRI slices with spontaneous hyperintensity in the precentral sulcus (yellow arrows), suggestive of cSAH. The same sulcus is relatively hyperintense in T2* (panel (**e**)) and not clearly discernible in the T1 sequence (panel (**f**)). Panels (**g**,**h**) illustrate MR angiography on the axial and coronal plane, reconstructed with the MIP/MPR tool (the red arrows focus on the occlusion).

**Figure 3 jcm-13-04382-f003:**
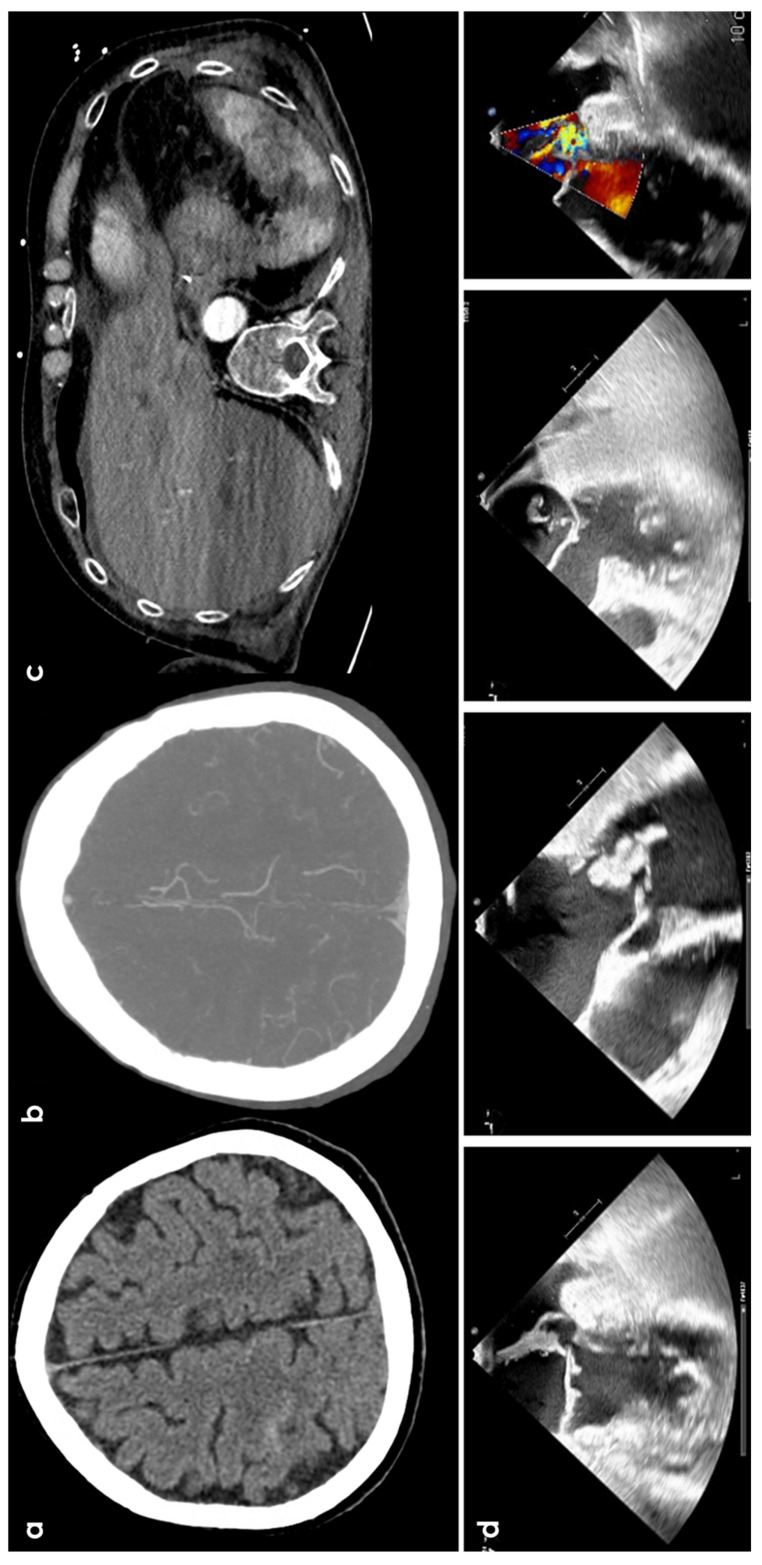
cSAH in a patient with infective endocarditis. Panel (**a**) shows NCCT with a linear hyperdense content in the right central sulcus. Panel (**b**) shows the corresponding axial reconstructed CT angiography with normal finding. Panel (**c**) underlines multiple splenic infarctions. Panel (**d**) shows multiple views of transesophageal echocardiography with a huge vegetation on the posterior leaf in the mitral valve.

**Figure 4 jcm-13-04382-f004:**
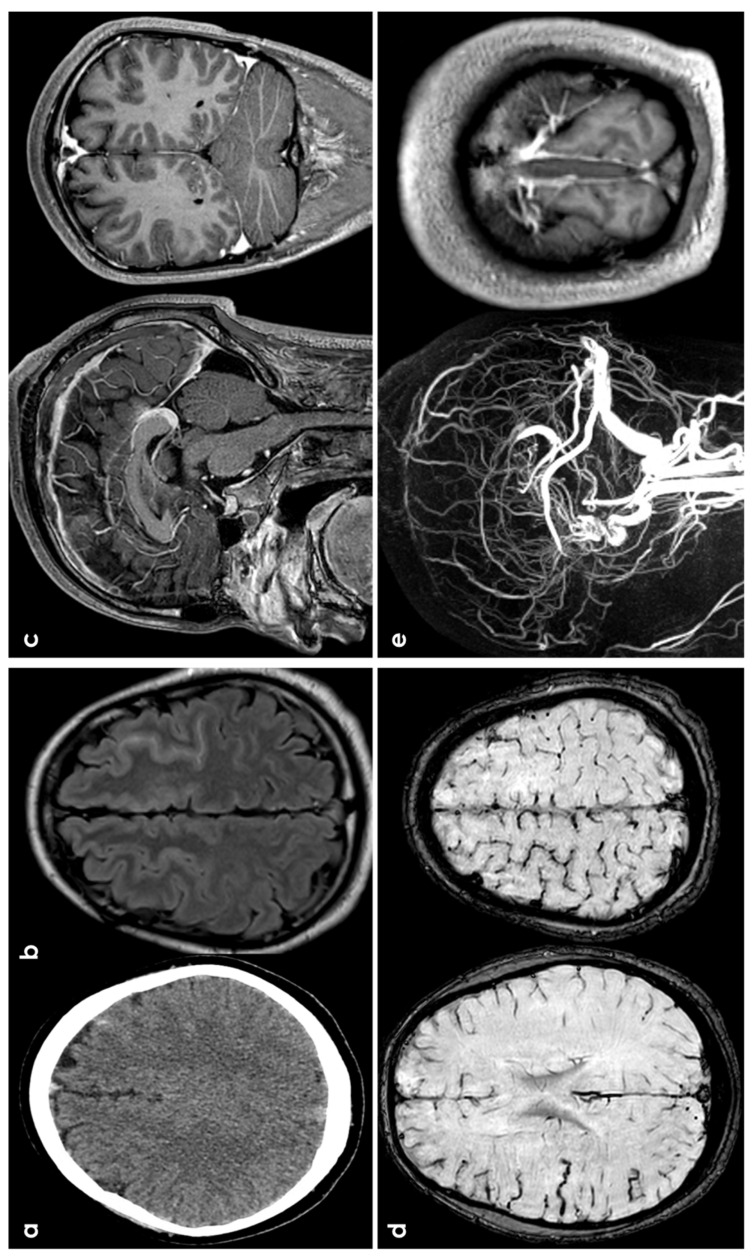
cSAH in a patient with CVT. Panels (**a**,**b**) show, respectively, NCCT and FLAIR MRI with, respectively, hyperdense and hyperintense left superior frontal sulcus, suggestive of SAH. Panel (**c**) shows sagittal and coronal view of post GBCA with an extensive superior sagittal sinus thrombosis. Panel (**d**) highlights in SWI the cortical venous congestion in the right hemisphere. Panel (**e**) focuses on MR angiography reconstructed in sagittal plane, confirming the SSS thrombosis and the lack of intravascular contrast in the posterior third of the SSS on T1-MRI.

**Figure 5 jcm-13-04382-f005:**
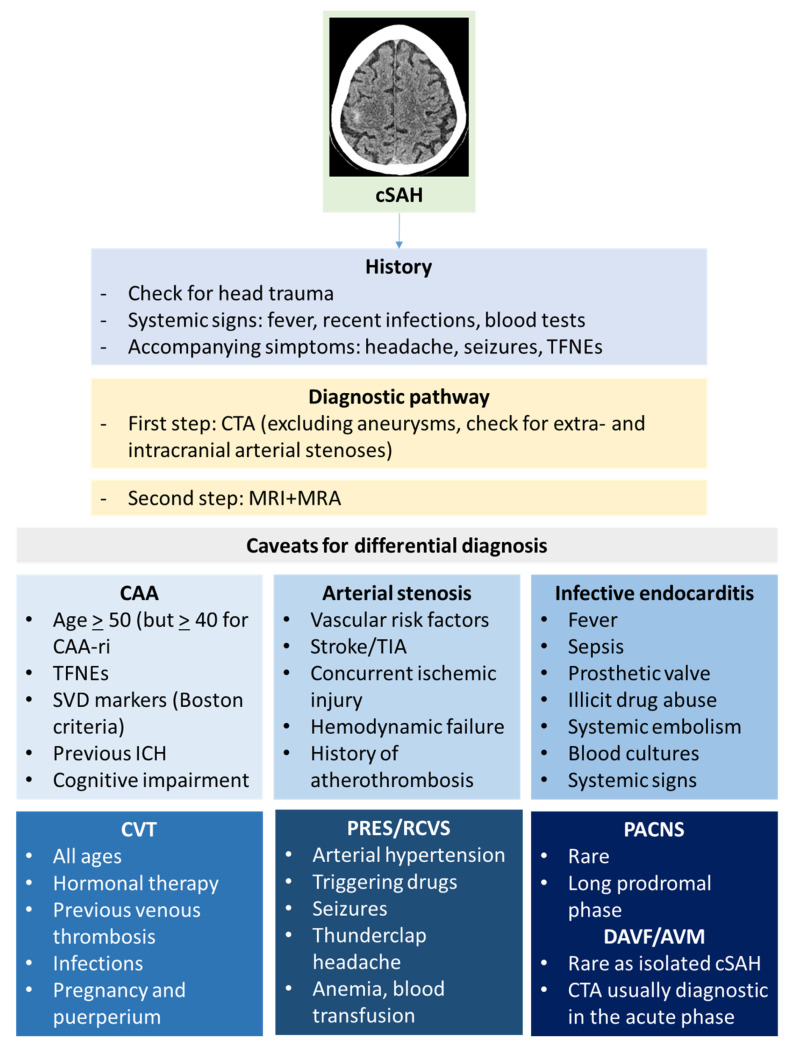
Diagnostic clues for cSAH.

## Data Availability

Not applicable.

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
