# Peer review of "Spontaneous Non-Aneurysmal Convexity Subarachnoid Hemorrhage: A Scoping Review of Different Etiologies beyond Cerebral Amyloid Angiopathy"

_jcm, 2024, doi:10.3390/jcm13154382_

Round 1
Reviewer 1 Report
Comments and Suggestions for Authors
This review is a very comprehensive summary of spontaneous non-aneurysmal convexity subarachnoid hemorrhage. If you drew a cartoon for each type of subarachnoid hemorrhage such as CAA, hemodynamic arterial stenosis. And as for infective endocarditis, you should show microaneurysms by angiography.
Author Response
We would thank the reviewer for his/her appreciation. We tried to draw a table with the features of the several caused of cSAH, but we were not satisfied from it in the first version of the review, so we added panles in order to describe different causes. The review targets cSAH and not aneurysmal SAH. This is the reson why we did not address the infectious aneurysms topic in infective endocarditis,
Reviewer 2 Report
Comments and Suggestions for Authors
I am pleased to be able to review the paper titled "Spontaneous non-aneurysmal convexity subarachnoid hemorrhage: a journey through different etiologies beyond cerebral amyloid angiopathy." by Zedde et al.
Overall, the manuscript is well-written and formatted for the journal. I am suggesting minor edits for the authors to consider.
1) Please revise the title if you agree to add a word or two about the type of review: For example, A narrative review, an Opinion article, or a rapid or scoping review. I don't think this was a systematic review.
2) Please add a few sentences at the end of the introduction regarding the methodology used to screen the articles and include them in this paper. Close to 100 references are cited, which took a lot of effort. Did the authors consider a systematic review or consult a librarian to ensure that key references are not missed?
3) May I suggest that the authors consider a clinical diagnostic workup workflow, a process map, or an algorithm that the readers may find helpful? This may enhance the appeal of your paper. I am particularly impressed by the neuro-imaging figures. Thus, either an algorithm or a table would complement the figures and may enhance the "journey," as the authors describe in the title.
4) The section regarding infective endocarditis may confuse readers as the paper's title is non-aneurysmal SAH, and the IE section details mycotic or septic aneurysms. Perhaps the authors can preface this in the introduction and emphasize whether IE-associated cSAH could be non-aneurysmal in etiology.
5) The clinical decision-making figure may follow Section 4.
6) Can you add a sentence or a paragraph regarding using Transcranial Doppler Ultrasound to screen for vasospasm? Are patients with cSAH more or less likely to experience cerebral vasospasm?
7) It may be helpful to add a table comparing aneurysmal SAH to c-SAH. Such a table may flush out the concepts of diagnosis, interventions, and complications. Along with the imaging, the clinical decision figure and the table should make your paper highly read and cited.
8) Please comment on the use of nimodipine in patients with cSAH. Readers may be curious. Please comment on the admission of patients with cSAH in an ICU. This may be a good point to consider in the comparison table.
Author Response
First of all, we would like to thank the refviewe for his/her comments and suggestions.
- We changed the title as suggested
- Thanks for the suggestion. We added the requested information.
- Thanks for the suggestion, We added a figure with the main diagnostic clues
- Good observation. We highlited this information
- We added it in the section summarizing the diagnostic pathway
- cSAH is not usually associate to vasospasm (apart RCVS but in tjhis case vasospasm precedes SAH and it may be present without cSAH) and the monitoring of vasospasm was considered out of topic in this case
- We decided to completely skip aneurysmal SAH after CTA excluding aneurysms as first step in diagnostic pathway of cSAH
- In a similar way we did not address treatment at all, but if the Ediro agrees we could add some notes. Our main concern is that the review is already long only for diagnostic issues.
Reviewer 3 Report
Comments and Suggestions for Authors
The review focus at a vastly unrecognized type of seemingly rare and “minor” cerebral haemorrhagic stroke: a subarachnoid bleeding of non-aneurysm origin. This condition often escapes proper and exact diagnosis due to its volatile nature. At the same time the diagnosis is important because the bleeding may herald a more serious condition and treatment may not only vary but be quite opposite depending on the real nature of the pathological process. For the above reasons the topic of the review is important and the authors addressed an important clinical challenge. The study's strengths include its detailed insight into the clinical background of the condition and clear diagnostic guidance. It is quite interesting as a summary of knowledge for a clinician - neurologist
There are some points that need to be addressed before publishing can be decided.
The narrative of this review is long-winded and sometimes undisciplined. A significant part of the information quoted in the article is basically textbook knowledge and could be curtailed and/or referred to appropriate neurological book chapters. Generally, this review itself looks more as a book chapter rather than as a sound scientific report. I would suggest using PRISMA methodology to help in a more sharp focus on those problems which are currently under discussion , while curtailing the basic, well established data and opinions.
Below are examples of somewhat messy narration of the article.
Some sentences/statements empty of meaning, even tautological, like line 72: “The Amyloid β (Aβ) deposition in CAA probably results from an imbalance between the production and clearance of Aβ, a breakdown product of Aβ precursor protein”.
Many awkward sentences, difficult to follow, like line 66: “CAA is a relevant cause of cSAH because the underlying disease has implication for the lifetime and not only for the individual cSAH episodes, which often recover without residual symptoms”. Or the whole paragraph lines 292-297: “Valvular vegetations and their length were significantly associated with cSAH (p<0.0001). Other lesions, such as perivalvular abscess, DWILs, brain macro-hemorrhages, intracranial co-infections, and visceral emboli, showed no significant difference between cSAH and non-cSAH groups. CMBs were equally observed in both groups. cSAH was unrelated to DWILs or CMBs. Two patients were excluded due to cSAH near cortical microinfarcts. Microinfarcts were most frequent (70.6%), with territorial ischemia in 29.4% 297 of cases [38]”. To which data pertains
The authors use a huge number of acronyms, many of them unexplained or explained with the latest use in the text (like i.e. NCCT). This makes the article difficult to follow. Please provide an abbreviation section and consider avoiding acronyms wherever possible.
Also captions of figures ought to be self-explanatory and contain no undefined acronyms.
The formula of the review work is sometimes blurred: the authors in several moments refer to (own?) results with no reference. For example lines 118-120:” Nonetheless, our findings align with two other reports highlighting TIA-like symptoms as a common presentation of cSAH”. Also similar lines 118-119: “Nonetheless, our findings align with two other reports highlighting TIA-like symptoms as a common presentation of cSAH”. Our findings in a review not supported with a publication? Item lines 279-282.
There are also several grammatic and typographic errors, like lines 44, 87, 101, 231, 282
I would suggest rethinking the formula of the review, defining a target audience, trying more concise and target-oriented approach to the topic. Language edition seems also necessary.
Comments on the Quality of English LanguageEnglish seems to need improvement.
Author Response
First of alla, we would like to thank the reviewer for his/her observations. We agree about the narrative features of our review. We tried to target a broad audeince and we did not follow a formal PRISMA approach for this reason. One of the reviewers asked for a broader audience. Then we will wait the feedback from the Editor about potential heavy changes of the structure of the review.
We addressed all the signaled points (acronysms, grammar arrors, sentences hard to understand and so on). Unfortunately, the format of the review does not include a list of acronyms, but we tried to repeat them session by session.
Round 2
Reviewer 3 Report
Comments and Suggestions for Authors
Line 445, lateral (?) sinus. Please specify which sinus you mean.
Author Response
Many thanks for your appreciation,
I changed lateral sinus in transverse and sygmoid sinus.